# Effects of Dietary Supplementation of Alpha-Ketoglutarate in a Low-Protein Diet on Fatty Acid Composition and Lipid Metabolism Related Gene Expression in Muscles of Growing Pigs

**DOI:** 10.3390/ani9100838

**Published:** 2019-10-21

**Authors:** Jiashun Chen, Haihan Zhang, Hu Gao, Baoju Kang, Fengming Chen, Yinghui Li, Chenxing Fu, Kang Yao

**Affiliations:** 1College of Animal Science and Technology, Hunan Agricultural University, Changsha 410128, China; chenjiashun1988@163.com (J.C.); zhhous@163.com (H.Z.); gaohu_20190008@163.com (H.G.); baoju2019@126.com (B.K.); cfming@stu.hunau.edu.cn (F.C.); 2Institute of Subtropical Agriculture, Chinese Academy of Sciences, Changsha 410125, China; yaokang@isa.ac.cn

**Keywords:** alpha-ketoglutarate, growing pigs, fatty acid composition, intramuscular fat, lipid metabolism

## Abstract

**Simple Summary:**

Alpha-ketoglutarate (AKG) is a critical intermediate in the tricarboxylic acid cycle. AKG has been reported to participate in energy production, promote protein synthesis, and improve amino acid metabolism. However, whether AKG functionally participates in the regulation of fat metabolism remains unknown. The objective of this experiment was to evaluate the impact of dietary supplementation with AKG on lipid metabolism in a pig model. The present results suggest that AKG supplementation in a reduced-protein diet could increase the intramuscular fat (IMF) and monounsaturated fatty acid (MUFA) contents in the biceps femoris muscles of pigs. These effects could be linked to the altered lipid metabolism related gene mRNA expression, which promotes the absorption and deposition of fatty acids in the muscle tissues. The results of this study can provide better understanding of the mechanisms by which dietary AKG modulates muscle lipid metabolism in pigs, and this could help to improve pig feeding strategies and supply high-quality pork for humans.

**Abstract:**

The aim of the current study was to investigate whether dietary supplementation with alpha-ketoglutarate (AKG) in a reduced crude protein (CP) diet would affect fatty acid composition and lipid metabolism related gene expression in the muscles of growing pigs. A total of 27 Large White × Landrace growing pigs at 44 ± 1 d of age (11.96 ± 0.18 kg) were randomly allocated to three treatments (*n* = 9). Dietary treatments included: (1) normal protein diet with 20% crude protein (CP) (NP); (2) a low crude protein diet formulated to contain approximately 17% CP (LP); and (3) a low crude protein diet with 17% CP supplemented with 1% AKG at the expense of regular corn components (ALP). The experimental trial lasted 35 d. The results showed that compared with the NP and LP diets, supplementation with AKG in a low-protein diet increased the intramuscular fat (IMF), oleic acid (C18:1n-9), and monounsaturated fatty acid (MUFA) contents (*p* < 0.05), and tended to increase the percentage of palmitoleic acid (C16:1) and stearic acid (C18:0) (*p* < 0.10) in the biceps femoris and longissimus dorsi muscles of growing pigs. These effects may be associated with increased relative mRNA expression levels of fatty acid synthase (FAS), acetyl-CoA carboxylase (ACC), adipocyte determination and differentiation factor 1 (ADD1), fatty acid binding protein 4 (FABP4), and stearoyl-CoA desaturase (SCD) in skeletal muscle, indicating that AKG might be involved in the differential regulation of some key lipogenic genes in skeletal muscles of pigs.

## 1. Introduction

With the increasing focus on the quality of pork for human consumption, fatty acid composition has been investigated as a desirable attribute in muscle and adipose tissues of farm animals [1]. The quality of a pig carcass and cuts is mostly dependent on the muscle and fat contents [2]. Increasing skeletal muscle growth and reducing excess fat accretion are major goals for pig production [3]. Intramuscular fat (IMF), also termed marbling fat, which is the total lipid within the skeletal muscles, plays a prominent role in meat quality, and the content of IMF is directly correlated with the quality of flesh including the nutritional value, flavor, and texture [4]. Nutritional modulation is an efficient way to change meat quality by affecting the muscle/fat ratio and composition [5,6,7]; however, the underlying mechanisms are still largely unknown.

Alpha-ketoglutarate (AKG) is a molecule that has a central role in the Krebs cycle, determining the overall rate of the citric acid cycle of the organism, and is also a nitrogen scavenger in the body. AKG can be rapidly converted into glutamate through the transamination by glutamate dehydrogenase, and further into glutamine through the amination by glutamine synthase [8]. AKG has a great effect on suppressing the production of oxygen free radicals and preventing peroxidative damage of lipids by participating in nonenzymatic oxidative decarboxylation during the decomposition of hydrogen peroxide [9]. Infusion of AKG may contribute to improved nutrient utilization and energy expenditure in growing pigs [10]. Furthermore, dietary supplementation with AKG improves energy status by modulating the AMP-activated protein kinase (AMPK) signaling pathway in the small intestine of piglets [11]. The addition of AKG can also enhance energy status by activating mechanistic target of rapamycin (mTOR) signaling, which inhibits acetyl-coenzyme A carboxylase (ACC) β in skeletal muscles of piglets [12]. Similarly, dietary AKG supplementation could increase muscle mass of weanling piglets [13]. Yao et al. (2013) reported that mTOR pathway was induced by AKG supplementation, which increased skeletal muscle mass in rats [14]. Our previous study demonstrated that the addition of AKG to a low-protein diet improved growth performance [15] and promoted the cells in skeletal muscles to synthesize amino acids in growing pigs [16]. Additionally, the muscular fiber and fatty acid composition in pig skeletal muscles were modulated by the dietary supplementation of monosodium l-glutamate [3]. These studies revealed that AKG might also affect muscle flesh quality and lipid metabolism in pigs, which is valuable to investigate. Several studies have reported that synthetic amino acid plays a critical role in lipid metabolism of pigs [6,17]. However, to the best of our knowledge, there are no studies on the effects of dietary AKG supplementation on lipid metabolism in pigs. Therefore, this study was conducted to investigate whether dietary supplementation with AKG in a reduced protein diet would affect fatty acid composition and lipid metabolism related gene expression in the muscles of growing pigs.

## 2. Materials and Methods

All experimental procedures used throughout this study were approved by the Committee of Animal Ethics at Hunan Agricultural University (permit number: CACAHU 2018-0069). The AKG (purity ≥98%) was obtained from Hubei Yuancheng Saichuang Technology Co., Ltd. (Wuhan, China).

### 2.1. Animals, Experimental Design, and Diets

A total of 27 Large White × Landrace growing pigs at 44 ± 1 d of age (11.96 ± 0.18 kg) were randomly allocated to the 3 following treatment groups: (1) normal protein diet with 20% crude protein (CP) (NP); (2) a low crude protein diet formulated to contain approximately 17% CP (LP); and (3) a low crude protein with 17% CP supplemented with 1% AKG at the expense of regular corn components (ALP). There were 9 replicates in each treatment, with 1 pig per replicate. The 3 diets were formulated based on corn–soybean meal to be isocaloric and meet the nutritional needs of these animals following National Research Council (NRC, 2012) guidelines (Table 1) [18]. Pigs were housed individually in cages equipped with a feeder and a nipple drinker, and all pigs had *ad libitum* access to clean drinking water and their assigned diets. The experiment lasted 35 d.

### 2.2. Sample Collection and Preparation

All pigs were weighed individually at the beginning and the end of the experiment. The feed consumption was recorded at the end of the experiment to calculate average daily gain, average daily feed intake, and feed-to-gain ratio, as previously described [15]. When the feeding experiment ended, 8 mL blood samples were collected via jugular vein puncture into 10 mL tubes. The samples were centrifuged at 3000× *g* and 4 °C for 10 min to separate out the serum and were stored at −20 °C until analysis. Subsequently, the pigs were euthanized after being anesthetized by intraperitoneal injection of sodium pentobarbital. Samples of the longissimus dorsi and biceps femoris muscles were collected immediately, followed by sharp freezing in liquid nitrogen and −80 °C storage.

### 2.3. Measurement of Serum Lipid Related Substances

Serum nonesterified fatty acids (NEFAs), triglyceride, low-density lipoprotein cholesterol (LDL-C), high-density lipoprotein cholesterol (HDL-C), and total cholesterol (TC) concentrations in serum were assayed using a biochemical analytical instrument (Cobas^®^ c311, Basel, Switzerland).

### 2.4. Muscle Fatty Acid Composition

Total lipid and fatty acid composition in the muscle was measured according to the procedure of Chen et al. [19]. The detailed gas chromatography (GC) parameters were as described previously [20]. The concentration of individual fatty acids was quantified according peak area and expressed as a percentage of total fatty acids.

### 2.5. RNA Extraction, Complementary DNA Synthesis, and Quantitative RT-PCR

Total RNA isolation from longissimus dorsi and biceps femoris muscle tissues using Trizol reagent (Invitrogen, Carlsbad, CA, USA) was performed according to the manufacturer’s instructions. Purified and quality extracted RNA was carried out, as previously described [16].

Quantitative real-time PCR (qRT-PCR) was performed in triplicate for each complementary DNA (cDNA) sample, using an SYBR^®^ Premix Ex Taq™ II qPCR kit (Takara Biotechnology Co. Ltd., Dalian, China) according to the manufacturer’s guidelines on an ABI7900HT real-time PCR system (Applied Biosystems, Forest City, CA, USA). Amplification was performed with the following conditions: denaturation for 10 min at 95 °C, followed by 40 PCR cycles of denaturation for 15 s at 95 °C, and annealing and extension for 60 s at 56–64 °C. Gene expression was normalized to GAPDH (internal reference) and presented as relative fold change compared with control (NP). All samples were measured in triplicate. The mRNA expression levels of target genes were calculated using the 2^−ΔΔCt^ method [21]. All PCR primers used in this study are listed in Table 2 [22,23].

### 2.6. Statistical Analysis

Data collected in the present study were analyzed by analysis of variance, using the general linear model (GLM) procedures of SPSS v. 20.0 (IBM, Armonk, NY, USA). Differences between treatment means were classified using Tukey’s multiple comparison test. Results are presented as mean ± standard error. For all statistical analyses, significance level was set at *p* < 0.05, and tendency was declared at 0.05 < *p* < 0.10.

## 3. Results

### 3.1. Serum Biochemical Parameters

Pigs fed the ALP diet had lower (*p* < 0.05) circulating concentrations of TC and triglyceride compared with those fed NP and LP diets (*p* < 0.05) (Table 3). The serum concentrations of LDL cholesterol in the ALP group tended to be lower (*p* < 0.10). However, pigs fed the ALP diet had higher concentrations of HDL cholesterol than those fed the NP diet (*p* < 0.05). No differences (*p* > 0.10) in serum concentrations of nonesterified fatty acids were observed among the NP, LP, and ALP treatments.

### 3.2. Intramuscular Fat and Fatty Acid Profile of Muscle

The IMF content of biceps femoris muscle with the LP and ALP diets was higher than that with the NP diet (*p* < 0.05) (Figure 1). No differences (*p* > 0.10) in the IMF content of longissimus dorsi muscle were found among the NP, LP, and ALP treatments.

In comparison to the NP and LP diets, pigs fed low-protein diets supplemented with AKG tended to have an increased percentage of palmitoleic acid (C16:1) of the longissimus dorsi muscle, but decreased percentage of C17:0 (*p* < 0.10) (Table 4). Compared with the NP diet, the ALP diet significantly increased the percentage of monounsaturated fatty acid (MUFA) of the longissimus dorsi muscle in growing pigs (*p* < 0.05).

Compared with the NP and LP diets, the ALP diet showed significantly greater oleic acid (C18:1n-9) and MUFA (*p* < 0.05) content and an increased n-6:n-3 PUFA ratio (*p* < 0.05), and tended to increase the percentage of C16:1and stearic acid (C18:0) (*p* < 0.10) (Table 5). In addition, compared with the NP diet, pigs fed the ALP diet had a decreased percentage of eicosatrienoic acid (C20:3n-6) (*p* < 0.05).

### 3.3. Gene mRNA Expression Levels in Muscle

The mRNA expression levels of key genes involved in lipid metabolism were quantified to determine whether the genes are regulated by supplementation of low-protein diets with AKG. Figure 2 shows the mRNA expression levels of genes related to lipogenesis (acetyl-CoA carboxylase, ACC; stearoyl-CoA desaturase, SCD; adipocyte determination and differentiation factor 1, ADD1; fatty acid synthase, FAS); lipolysis (hormone-sensitive lipase, HSL; lipoprotein lipase, LPL); and lipid transport (fatty acid binding protein 4, FABP4) in the longissimus dorsi muscle (Figure 2a) and biceps femoris muscle (Figure 2b).

In the longissimus dorsi muscle, there was no difference in mRNA levels of ACC, FAS, LPL, SCD, FABP4, and HSL among the groups (*P* > 0.05). The ALP group showed a higher level of mRNA for ADD1 than the other two groups (*P* < 0.05).

In the biceps femoris muscle, there was no difference in mRNA levels of LPL among the groups (*P* > 0.05). The ALP diet increased the mRNA levels of FAS, ACC, ADD1, FABP4, and SCD (*P* < 0.05), but decreased the mRNA level of HSL when compared with the NP and LP diets (*P* < 0.05). However, no significant difference was found between the NP and LP diets (*P* > 0.05).

## 4. Discussion

In this study, the primary aim was to investigate the IMF content and fatty acid profile in muscle tissues of growing pigs that were fed with a reduced protein diet supplemented with AKG. The secondary aim was to investigate the effects of supplementation with AKG on the mRNA expression levels of genes that are involved in lipid metabolism.

Over the last decades, commercial pig lines have benefited from intensive artificial selection to reduce the subcutaneous fat content, but this brought a concomitant decrease in marbling or IMF [23]. The IMF content is one of the most important traits that determines the characteristics of meat quality, such as tenderness, juiciness, and flavor [24]. It has been found that a higher IMF percentage in pork, including more monounsaturated and less polyunsaturated fatty acid, could be achieved by reducing the protein content of the pig diet [25,26]. Ranges of dietary protein concentrations (e.g., 21% vs. 18% and 20% vs. 17%) have been demonstrated to increase the IMF content in crossbred pigs [27,28]. In this study, the IMF content in muscle was also confirmed to be increased by feeding with a low-protein diet. To our knowledge, this is the first time an in vivo approach was used to investigate increased IMF in pigs fed low-protein diets supplemented with AKG, and our results on the effects of AKG supplementation are consistent with those of Tan et al. [29] and Madeira et al. [23], who demonstrated that dietary supplementation with 1% arginine increased IMF content in pig muscle. Therefore, a reduced-protein diet supplemented with AKG changed muscle composition, which might be related to alteration of lipid metabolism in muscle. Additionally, it has been reported that energy-related phosphorylation of AMPK and energy oxidation were stimulated by AKG in the intestinal mucosa [11]. The subsequent inactivation of the downstream target enzyme ACC, which is responsible for converting acetyl-CoA into malonyl-CoA, an inhibitor of fatty acid synthesis, and enhancing ATP supply to promote fat storage, improved the energy status in the skeletal muscles of pigs [12].

In recent years, the fatty acid composition of meat has been reported to be altered due to changes in target diets during the animal production cycle [7,25,26]. Therefore, the balance between fatty acid deposition and absorption could be indicated by the variations of fatty acid profile that is detected in skeletal muscle [22]. Fatty acid composition is also an important factor that determines the nutritional value and oxidative stability of muscle [27]. Since saturated fatty acid (SFA) and monounsaturated fatty acid (MUFA) are positively correlated with meat flavor [30], they are considered as essential indices of meat quality. Nutritional recommendations for a healthy swine diet suggest that the ratio of PUFA to SFA (PUFA/SFA) should be at least 0.40 [31]. A higher ratio of PUFA to SFA was found to be associated with a lower risk of coronary heart disease (CHD) in humans [32]. In this study, low crude protein diets supplemented with AKG increased the contents of palmitoleic acid (C16:1), oleic acid (C18:1n-9), and MUFA in skeletal muscles of pigs. Furthermore, the ratio of each kind of PUFA to SFA was over 0.40. These results might suggest that the improved capacity of PUFA synthesis in pigs and the change of the fatty acid profile in pig muscle might result from the AKG supplementation, which could be involved in MUFA synthesis in muscle, in a protein-reduced diet.

On the other hand, it has been reported that AKG can be converted into leucine, arginine, proline, and other amino acids in enterocytes [33]. Amino acids (e.g., leucine, isoleucine, arginine) play important roles in energy homeostasis and adipogenesis [17,22,34]. Thus, the alteration of IMF and MUFA content in muscle caused by dietary supplementation of AKG could be associated with the ectopic expression of genes involved in lipid metabolism that modulates muscle lipogenesis. SCD, the rate-limiting enzyme in the biosynthesis of MUFA from SFA, could catalyze the conversion of palmitate and stearate to the corresponding unsaturated fatty acids (C16:1, C18:1), which are important substrates for the synthesis of cholesterol and phospholipids [35,36]. In our results, pigs in the ALP group showed upregulation of SCD mRNA expression in biceps femoris muscle, which corresponded to the increased MUFA content in skeletal muscle and might indicate the induction of lipogenesis in IMF in pigs that were fed with an AKG-supplemented low-protein diet.

The indispensable rate-limiting enzymes in de novo lipogenesis (DNL) are fatty acid synthase and DNL commitment enzyme ACC [37]. ADD1 has been regarded as a principal regulatory transcription factor that controls the expression of PPARγ gene to promote DNL in animals [38,39]. FAS was found to be a key regulatory enzyme in fatty acid synthesis, which is involved in the final step of fatty acid biosynthesis in tissues [40], while HSL, as the most important lipase for intracellular triglyceride and diglyceride hydrolysis, catalyzed the first two degradation procedures of triglycerides and diglycerides [41]. Additionally, HSL can also facilitate the absorption and deposition of lipids [42]. FABP4, as a member of the fatty acid binding protein family, is responsible for the transport of fatty acids in adipocytes [43]. In this study, we did not identify any significant effect of a low-protein diet on the mRNA expression levels of key genes such as ACC, FAS, FABP4, and LPL controlling fatty acid deposition in both muscle tissues compared with control diet. Such findings are consistent with the results published by Madeira et al. [44]. Notably, our results reveal that feeding with a low-protein diet supplemented with AKG boosted the transcriptional expression of ACC, ADD1, FAS, and FABP4, but reduced the expression level of HSL in biceps femoris muscle. This might indicate that superior fatty acid transport and synthesis occurred, leading to the higher IMF content and fatty acid availability in biceps femoris muscle of pigs in the ALP group. Thus, it can be speculated that AKG supplementation in a low-protein diet may elevate the influx of fatty acids in different locations of the skeletal muscle, particularly in the biceps femoris muscle in pigs, but the underlying mechanism is still unknown. In order to further improve meat quality as well as nutritional value, it is still necessary to explore its underlying mechanism in detail.

## 5. Conclusions

In conclusion, the results obtained under the conditions of this experiment suggest that supplementation with AKG in a low-protein diet could increase the IMF and MUFA content in biceps femoris muscle of pigs. This might be a result of the alteration of mRNA expression level of important genes involved in lipid metabolism, further causing a change of absorption and utilization of fatty acids in muscle tissue. These changes could also provide a better understanding of how dietary AKG regulates muscle lipogenesis in pigs, and could potentially help to improve pig feeding strategies and supply high-quality pork for humans.

## Figures and Tables

**Figure 1 animals-09-00838-f001:**
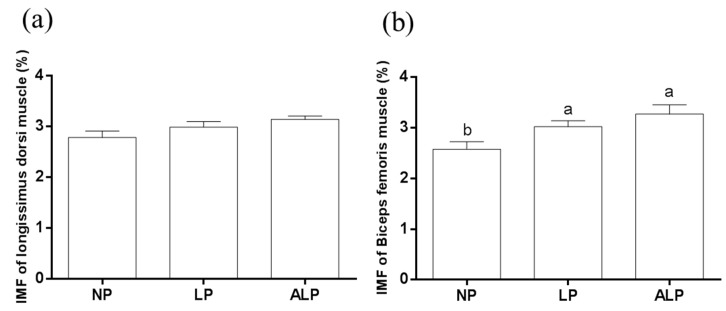
Effects of dietary AKG supplementation of a low crude protein diet on intramuscular fat (IMF) in the longissimus dorsi (**a**) and biceps femoris (**b**) muscles of growing pigs. Results are means ± SEM, representing nine animals per treatment. Bars without a common letter differ significantly (*p* < 0.05). NP, normal protein; LP, low crude protein; ALP, AKG + low crude protein.

**Figure 2 animals-09-00838-f002:**
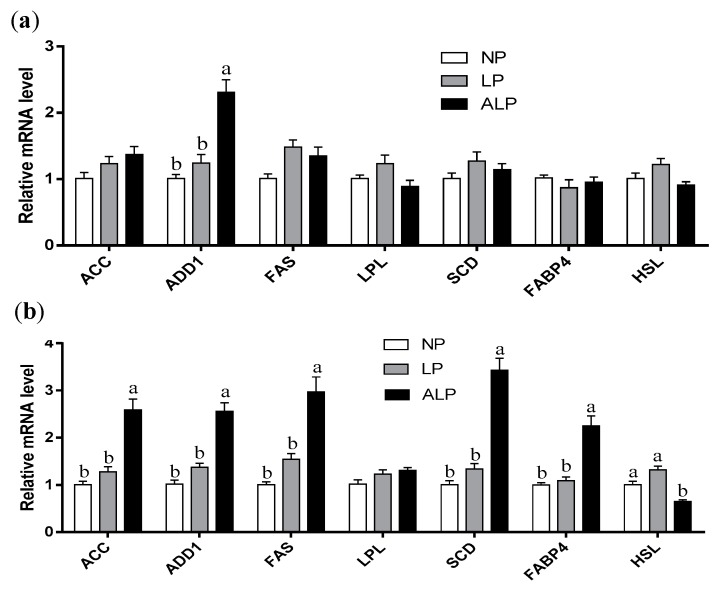
Effects of dietary AKG supplementation of a low-protein diet on the relative mRNA expression levels of genes in the longissimus dorsi muscle (**a**) and biceps femoris muscle (**b**) of the growing pigs. Results are means ± SEM, representing nine animals per treatment. Bars without a common letter differ significantly (*p* < 0.05). NP, normal protein; LP, low crude protein; ALP, AKG + low crude protein.

**Table 1 animals-09-00838-t001:** Ingredient composition and nutrient levels in experimental diets (as-fed basis, %).

Items	Dietary Treatment ^a^		
	NP	LP	ALP
Ingredient (%)			
Corn	63.64	66.50	65.50
Soybean meal	19.80	18.80	18.80
Dried whey	4.30	4.30	4.30
Fish meal	9.00	4.00	4.00
Soybean oil	0.80	2.60	2.60
AKG ^b^	0.00	0.00	1.00
Limestone	0.50	0.60	0.60
Monocalcium phosphate	0.00	0.74	0.74
L-lysine-HCl	0.41	0.65	0.65
L-threonine	0.11	0.25	0.25
DL-methionine	0.13	0.20	0.20
L-tryptophan	0.01	0.06	0.06
Sodium chloride	0.30	0.30	0.30
Premix ^c^	1.00	1.00	1.00
Total	100.00	100.00	100.00
Nutrient level (%)			
Digestible energy (MJ/kg) ^d^	14.60	14.60	14.60
Crude protein	20.19	17.21	17.20
Lysine	1.25	1.24	1.23
Methionine	0.38	0.37	0.37
Methionine + cysteine	0.62	0.65	0.63
Threonine	0.76	0.73	0.74
Tryptophan	0.22	0.22	0.21
Total calcium	0.72	0.72	0.71
Total phosphorus	0.65	0.64	0.63

^a^ NP, normal protein; LP, low crude protein; ALP, AKG + low crude protein. ^b^ AKG, alpha-ketoglutarate. ^c^ Supplied per kg of diet: CuSO_4_·5H_2_O, 19.8 mg; KI, 0.20 mg; FeSO_4_·7H_2_O, 400 mg; NaSeO_3_, 0.56 mg; ZnSO_4_·7H_2_O, 359 mg; MnSO_4_ H_2_O, 10.2 mg; Vitamin K (menadione), 5 mg; Vitamin B_1_, 2 mg; Vitamin B_2_, 15 mg; Vitamin B_12_, 30 g; Vitamin A, 5400 IU; Vitamin D_3_, 110 IU; Vitamin E, 18 IU; Choline chloride, 80 mg; Antioxidants, 20 mg; Fungicide, 100 mg. ^d^ Calculated values, other values were measured directly.

**Table 2 animals-09-00838-t002:** Primers used for real-time quantitative PCR in this study.

Gene	Primer Sequence (5′-3′)	Size (bp)	Accession No.
ACC	F: GGCCATCAAGGACTTCAACC	120	NM_001114269
	R: ACGATGTAAGCGCCGAACTT		
ADD1	F: GCGACGGTGCCTCTGGTAGT	218	XM_021066226.1
	R: CGCAAGACGGCGGATTTA		
FAS	F: GCCTAACTCCTCGCTGCAAT	195	NM_001099930.1
	R: TCCTTGGAACCGTCTGTGTTC		
LPL	F: CTCGTGCTCAGATGCCCTAC	148	NM_214286.1
	R: GGCAGGGTGAAAGGGATGTT		
SCD	F: GCCTACTATCTGCTGAGTGC	152	XM_021072070.1
	R: TCTCGGGCCCATTCATAAAC		
FABP4	F: TGGAAACTTGTCTCCAGTG	147	NM_001002817.1
	R: GGTACTTTCTGATCTAATGGTG		
HSL	F: TCGGAGTGAACGGATTTG	195	-
	R: TCCTCCTTGGTGCTAATCTCGT		
GAPDH	F: TCGGAGTGAACGGATTTG	219	NM_001206359.1
	R: CCTGGAAGATGGTGATGG		

**Table 3 animals-09-00838-t003:** Serum lipid-related substances of growing pigs fed normal protein or low-protein diets without or supplemented with AKG.

Items	Dietary Treatment ^1^			SEM	*p*-Value
	NP	LP	ALP		
Non-esterified fatty acid, μmol/L	610.43	578.68	605.74	11.81	0.261
HDL-cholesterol, mmol/L	0.78 ^b^	0.84 ^a,b^	0.96 ^a^	0.12	0.032
LDL-cholesterol, mmol/L	1.47	1.36	1.32	0.03	0.074
Triglyceride, mmol/L	0.46 ^a^	0.38 ^a^	0.24 ^b^	0.04	0.028
TC, mmol/L	2.96 ^a^	2.88 ^a^	2.73 ^b^	0.05	0.039

^1^ NP, normal protein; LP, low crude protein; ALP, AKG + low crude protein. ^a,b^ Means within a row with different superscripts differ (*p* < 0.05).

**Table 4 animals-09-00838-t004:** Fatty acid composition (% of total fatty acids) of longsissimus dorsi muscle from growing pigs fed normal protein or low-protein diets without or supplemented with AKG.

Items	Dietary Treatment ^1^			SEM	*p*-Value
	NP	LP	ALP		
C12:0	0.06	0.07	0.06	0.02	0.265
C13:0	0.19	0.17	0.16	0.06	0.704
C14:0	0.93	0.90	0.84	0.11	0.586
C15:0	0.22	0.19	0.17	0.02	0.779
C15:1	0.16	0.16	0.15	0.03	0.925
C16:0	22.14	21.97	21.13	1.71	0.553
C16:1	1.89	1.84	2.29	0.20	0.095
C17:0	1.15	1.05	0.86	0.11	0.086
C18:0	12.44	11.61	11.34	0.26	0.812
C18:1n-9	30.13	30.99	31.77	2.21	0.115
C18:2n-6	16.33	16.04	15.59	1.13	0.269
C20:0	0.19	0.18	0.20	0.03	0.662
C18:3n-3	0.51	0.50	0.48	0.07	0.238
C20:2n-6	0.59	0.61	0.59	0.14	0.146
C20:3n-6	0.33	0.30	0.28	0.06	0.804
C20:3n-3	0.15	0.14	0.12	0.05	0.173
C20:4n-6	1.72	1.66	1.60	0.19	0.147
C20:5n-3	0.16	0.15	0.14	0.10	0.216
C22:6n-3	0.13	0.13	0.12	0.29	0.418
SFA ^2^	37.30	36.15	34.76	2.75	0.352
MUFA ^3^	32.17 ^b^	32.99 ^a,b^	34.21 ^a^	2.34	0.049
PUFA ^4^	19.91	19.54	18.91	1.07	0.526
ΣPUFA:SFA	0.53	0.54	0.54	0.09	0.599
Σn-3 PUFA ^5^	0.94	0.93	0.86	0.30	0.198
Σn-6 PUFA ^6^	18.97	18.61	18.05	1.17	0.371
Σn-6:n-3 PUFA	20.19	20.05	21.00	0.84	0.642

^1^ NP, normal protein; LP, low crude protein; ALP, AKG + low crude protein. ^2^ SFA = C12:0 + C13:0 + C14:0 + C15:0 + C16:0 + C17:0 + C18:0 + C20:0. ^3^ MUFA = C15:1 + C16:1 + C18:1n-9. ^4^ PUFA = C18:2n-6 + C18:3n-6 + C18:3n-3 + C20:3n-6 + C20:3n-3+ C20:4n-6 + C20:5n-3 + C22:6n-3. ^5^ n-3 PUFA = C18:3n-3 + C20:3n-3 + C20:5n-3+ C22:6n-3. ^6^ n-6 PUFA = C18:2n-6 + C18:3n-6 + C20:3n-6 + C20:4n-6. ^a,b^ Means within a row with different superscripts differ (*p* < 0.05).

**Table 5 animals-09-00838-t005:** Fatty acid composition (% of total fatty acids) of biceps femoris muscle of growing pigs fed normal protein or low-protein diets without or supplemented with AKG.

Items	Dietary Treatment ^1^			SEM	*p*-Value
	NP	LP	ALP		
C12:0	0.06	0.06	0.07	0.02	0.667
C13:0	0.26	0.29	0.22	0.09	0.460
C14:0	0.82	0.80	0.78	0.08	0.851
C15:0	0.22	0.23	0.30	0.08	0.636
C15:1	0.16	0.18	0.13	0.06	0.502
C16:0	21.36	20.75	20.52	0.60	0.563
C16:1	1.91	1.66	2.00	0.21	0.097
C17:0	1.01	1.08	0.98	0.36	0.495
C18:0	10.65	11.37	11.84	0.99	0.077
C18:1n-9	27.67 ^b^	28.54 ^b^	31.94 ^a^	0.65	0.046
C18:2n-6	19.62	19.61	18.38	0.39	0.119
C20:0	0.15	0.15	0.17	0.03	0.286
C18:3n-3	1.02	0.94	0.83	0.11	0.256
C20:2n-6	0.75	0.71	0.67	0.09	0.243
C20:3n-6	0.38 ^a^	0.35 ^a,b^	0.24 ^b^	0.13	0.041
C20:3n-3	0.15	0.14	0.13	0.05	0.203
C20:4n-6	2.40	2.27	2.11	0.13	0.477
C20:5n-3	0.24	0.23	0.20	0.05	0.219
C22:6n-3	0.18	0.17	0.17	0.02	0.312
SFA ^2^	34.54	34.74	34.89	2.30	0.165
MUFA ^3^	29.73 ^b^	30.38 ^b^	34.07 ^a^	1.26	0.038
PUFA ^4^	24.74	24.42	22.73	1.39	0.219
ΣPUFA:SFA	0.72	0.70	0.65	0.07	0.104
Σn-3 PUFA ^5^	1.58	1.48	1.33	0.03	0.135
Σn-6 PUFA ^6^	23.15	22.94	21.40	2.44	0.132
Σn-6:n-3 PUFA	14.61 ^b^	15.49 ^a,b^	16.08 ^a^	1.51	0.026

^1^ NP, normal protein; LP, low crude protein; ALP, AKG + low crude protein. ^2^ SFA = C12:0 + C13:0 + C14:0 + C15:0 + C16:0 + C17:0 + C18:0 + C20:0. ^3^ MUFA = C15:1 + C16:1 + C18:1n-9. ^4^ PUFA = C18:2n-6 + C18:3n-6 + C18:3n-3 + C20:3n-6 + C20:3n-3+ C20:4n-6 + C20:5n-3 + C22:6n-3. ^5^ n-3 PUFA = C18:3n-3 + C20:3n-3 + C20:5n-3+ C22:6n-3. ^6^ n-6 PUFA = C18:2n-6 + C18:3n-6 + C20:3n-6 + C20:4n-6. ^a,b^ Means within a row with different superscripts differ (*p* < 0.05).

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
