# Peer review of "Effects of Dietary Supplementation of Alpha-Ketoglutarate in a Low-Protein Diet on Fatty Acid Composition and Lipid Metabolism Related Gene Expression in Muscles of Growing Pigs"

_animals, 2019, doi:10.3390/ani9100838_

Round 1

Author Response

Response to Reviewer 1 Comments

Point 1: 
 Please see the attached file

Response 1: We agree with the view. We have made correction according to the reviewer’s comments. Revised portion are marked in red in the paper.

Reviewer 2 Report

Please discuss possible reasons for the different response in biceps femoris than in longisimus dorsi.

Citations should point to the authors that found the original results, not to mentions in other papers.

Author Response

Response to Reviewer 2 Comments

Point 1: Citation 2 did not study quality; they studied measurement techniques.

Response 1: We agree with the view. We have made correction according to the reviewer’s comments. We have added “Kouba, M.; Bonneau, M. Compared development of intermuscular and subcutaneous fat in carcass and primal cuts of growing pigs from 30 to 140 kg body weight. Meat Sci. 2009, 81, 270-274.” Revised portion are marked in red in the paper.

Point 2: Citation 3 did not study the goals of pig production.

Response 2: We agree with the view. We have made correction according to the reviewer’s comments. We have added“Wood, J.D.; Nute, G.R.;Richardson, R.I.; Whittington, F.M.; Southwood, O.; Plastow, G.; Mansbridge, R.; da Costa, N.; Chang, K.C. Effects of breed, diet and muscle on fat deposition and eating quality in pigs. Meat Sci. 2004, 67, 651-667.”  Revised portion are marked in red in the paper.

Point 3: Citation 4 studied gene expression, not sensory or nutritional quality.

Response 3: We agree with the view. We have made correction according to the reviewer’s comments. We have added “Fernandez, X.; Monin, G.; Talmant, A.; Mourot, J.; Lebret, B. Influence of intramuscular fat content on the quality of pig meat-1. Composition of the lipid fraction and sensory characteristics of m. longissimus lumborum. Meat Sci. 1999, 53, 59-65.” Revised portion are marked in red in the paper.

Point 4: Citation 23 does not support this statement.

Response 4: We agree with the view. We have made correction according to the reviewer’s comments. We have added “Franco, D.; Vazquez, J.A.; Lorenzo, J.M. Growth performance, carcass and meat quality of the Celta pig crossbred with Duroc and Landrance genotypes. Meat Sci. 2014, 96, 195-202.”Revised portion are marked in red in the paper.

Point 5: Do you mean all protein levels within this range increase IMF?

Response 5: We agree with the view. The reviewer gave us a good suggestion. In the previous study, many studies have been reported to reduce feed protein level could increase the IMF content in pigs.

Point 6: Please discuss possible reasons for the different response in biceps femoris than in longisimus dorsi.

Response 6: We agree with the view. The reviewer gave us a good suggestion. In fact, in this study, we have discussed this reasons. “the AKG supplementation in a low protein diet may elevate the influx of fatty acids in different locations of the skeletal muscle, especially in the biceps femoris muscle in growing pigs, but the underlying mechanisms is still unknown. In order to further improving meat quality, as well as the nutritional value, it is still of a great necessity to explore the detailed mechanism underlying it.”  Moreover, in response to this question, we intend to continue to study this question in depth in the future.

Point 7: Citations should point to the authors that found the original results, not to mentions in other papers.

Response 7: We agree with the view. We have made correction according to the reviewer’s comments. Revised portion are marked in red in the paper.

Round 2

Reviewer 1 Report

Please refer to the document.

Author Response

Response to Reviewer 1 Comments

Point 1: 
 Please refer to the document.

Response 1: We agree with the view. We have made correction according to the reviewer’s comments. Revised portion are marked in red in the paper.
